# Autoinflammation in Syndromic Hidradenitis Suppurativa: The Role of AIM2

**DOI:** 10.3390/vaccines11010162

**Published:** 2023-01-11

**Authors:** Chiara Moltrasio, Rachele Cagliani, Manuela Sironi, Mario Clerici, Chiara Pontremoli, Carlo Alberto Maronese, Paola Maura Tricarico, Sergio Crovella, Angelo Valerio Marzano

**Affiliations:** 1Dermatology Unit, Fondazione IRCCS Ca’ Granda Ospedale Maggiore Policlinico, 20122 Milan, Italy; 2Department of Medical Surgical and Health Sciences, University of Trieste, 34137 Trieste, Italy; 3Scientific Institute IRCCS E. MEDEA, Bioinformatics, 23842 Bosisio Parini, Italy; 4Department of Pathophysiology and Transplantation, Università degli Studi di Milano, 20122 Milan, Italy; 5Don C. Gnocchi Foundation ONLUS, IRCCS, 20122 Milan, Italy; 6Department of Advanced Diagnostics, Institute for Maternal and Child Health-IRCCS “Burlo Garofolo”, 34137 Trieste, Italy; 7Biological Science Program, Department of Biological and Environmental Sciences, College of Arts and Sciences, University of Qatar, Doha 2713, Qatar

**Keywords:** innate immunity, AIM2, autoinflammation, hidradenitis suppurativa, syndromic autoinflammatory skin diseases

## Abstract

Background: AIM2 is a key cytoplasmatic pathogen-sensor that detects foreign DNA from viruses and bacteria; it can also recognize damaged or anomalous presence of DNA, promoting inflammasome assembly and activation with the secretion of IL-1β, thus sustaining a chronic inflammatory state, potentially leading to the onset of autoinflammatory skin diseases. Given the implication of the IL-1β pathway in the pathogenesis of syndromic hidradenitis suppurativa (HS), an autoinflammatory immune-mediated skin condition, the potential involvement of AIM2 was investigated. Methods: Sequencing of the whole coding region of the *AIM2* gene, comprising 5′- and 3′ UTR and a region upstream of the first exon of ~800 bp was performed in twelve syndromic HS patients. Results: Six out of twelve syndromic HS patients carried a heterozygous variant c.−208 A ≥ C (rs41264459), located on the promoter region of the *AIM2* gene, with a minor allele frequency of 0.25, which is much higher than that reported in 1000 G and GnomAD (0.075 and 0.094, respectively). The same variant was found at a lower allelic frequency in sporadic HS and isolated pyoderma gangrenosum (PG) (0.125 and 0.065, respectively). Conclusion: Our data suggest that this variant might play a role in susceptibility to develop syndromic forms of HS but not to progress to sporadic HS and PG. Furthermore, epigenetic and/or somatic variations could affect AIM2 expression leading to different, context-dependent responses.

## 1. Introduction

The innate immune system features a set of germ-line–-encoded pattern recognition receptors (PRRs) capable of recognizing molecules commonly found in pathogens, the so-called Pathogen-Associated Molecular Patterns—PAMPs, or released by damaged cells, termed Damage-Associated Molecular Patterns—DAMPs [1]. Absent in melanoma-2 (AIM2)-like receptors (ALRs) are cytoplasmic PRRs [2] that promote the assembly and activation of the inflammasome, a multi-protein entity known as the first responder in the innate branch of the immune system. The inflammasome activates the cysteine protease caspase 1 and subsequently promotes an inflammatory cell signaling pathway with the production and release of inflammatory cytokines, mainly IL-1β and IL-18; ultimately, a form of programmed cell death (PCD), called pyroptosis, is induced [3]. The protein AIM2, encoded by the *AIM2* gene (1q23.1-q23.2), is a cytoplasmatic sensor that responds to pathogen-derived double-stranded DNA molecules in both the cytosol and the nucleus [4]; it displays an N-terminal pyrin domain (PYD) and one highly conserved hematopoietic interferon (IFN)-inducible nuclear protein with 200-amino-acid repeat (HIN200) domain at the C-terminal region with tightly packed oligonucleotide or oligosaccharide binding folds [5]. After assembly and activation, the AIM2 inflammasome has a protective role in infectious diseases but may also be harmful in several sterile inflammatory diseases. Indeed, following pathological stimulation by self-DNA, this nucleic acid-recognizing inflammasome can lead to autoinflammatory, autoimmune and immune-mediated diseases, including psoriasis [6], systemic lupus erythematous [7] and atopic dermatitis [8]. It has been postulated that its upregulation may play a dual role: a first line of defense against invading pathogens and, in the case of prolonged impairment of the skin barrier, an effector supporting a chronic loop of inflammation, potentially leading to disease [8]. Nonetheless, further studies are needed to fully understand the function of the AIM2 inflammasome in the skin and its role in promoting autoinflammatory skin conditions.

Hidradenitis suppurativa (HS) is a chronic-debilitating autoinflammatory skin disease of the pilosebaceous unit, which is usually sporadic and clinically characterized by inflammatory nodules, abscesses, and sinus tracts on apocrine sweat gland-bearing skin [9]. In a rare subset of patients, HS may co-occur with other immune-mediated inflammatory diseases or inherited conditions, a setting aptly termed “syndromic” HS [10], whose prototype is represented by the clinical triad of pyoderma gangrenosum (PG), acne, and suppurative hidradenitis (PASH), to date considered as the paradigm of autoinflammatory syndromes within the spectrum of neutrophilic dermatoses (ND) [11]. The other main autoinflammatory syndromes involving HS are: PASH with pyogenic arthritis, termed PAPASH; PG, acne, HS, and ankylosing spondylitis (PASS) and synovitis, acne, pustulosis, hyperostosis, and osteitis (SAPHO), in which HS represents a common dermatological manifestation [12].

The pathogenesis of HS and its syndromic forms is complex, resulting from an interaction between genetic alterations-mainly related to autoinflammation and keratinization pathways [13,14,15], and environmental/lifestyle factors. Furthermore, a dysregulated adaptive and innate immune system response leading to a chronic inflammatory state, with an inflammasome-driven predominance of IL-1β, is a pivotal feature of HS and related forms [16].

Given the implication of the IL-1β pathway in pathogenesis of HS and its syndromic forms, we conducted a single-gene sequencing study to investigate the potential involvement of AIM2 inflammasome in this spectrum of diseases.

## 2. Materials and Methods

### 2.1. Patients

Twelve unrelated Italian patients with syndromic HS, recruited between 2011 and 2022 and followed-up at the Dermatology Unit, Fondazione IRCCS Ca’ Granda Ospedale Maggiore Policlinico, Milan, Italy, were enrolled in this study. All patients signed written informed consent approved by the Area B Milan Ethics Committee (protocol no. 487_2020).

### 2.2. Gene Sequencing

Total DNA was extracted from peripheral blood using QIAamp DNA Blood Mini Kit (QIAGEN, Milan, Italy) according to the recommended protocol. All *AIM2* (NM_004833) exons, flanking intron regions and ~ 800 bp upstream of the first exon were amplified through polymerase chain reaction (PCR) using oligonucleotide primers designed according to the established laboratory protocol (forward and reverse primers are reported in Appendix A). The PCR reactions (35 cycles) were carried out with JumpStart REDAccuTaq DNA Polymerase (Merck, Milan, Italy) using annealing and extension temperatures of 58 and 68 °C, respectively. The PCR products were verified on agarose gel, purified using ExoSAP-IT enzyme mix (incubation at 37 °C for 15 min; enzyme deactivation at 80 °C for 15 min) and subjected to direct nucleotide sequencing using the Big Dye™ Terminator v3.1 cycle sequencing kit (Thermo Fisher Scientific Inc., Monza, Italy) following the manufacturer’s instructions. The sequencing was performed on ABI PRISM 3500 xL Dx Genetic Analyzer (Applied Biosystems) while the sequences were analyzed with the Sequencing Analysis Software version 5.4 (Thermo Fisher Scientific Inc., Monza, Italy) and examined using the Nucleotide BLAST program.

The allele frequencies of the polymorphic variants identified in syndromic HS and in isolated HS and PG patients were compared with the frequencies of the minor allele frequency (MAF) reported in the 1000 Genomes Project (1000 G) [17], and GnomAD [18] databases for subjects of European ancestry.

## 3. Results

### 3.1. Clinical Features

Of the twelve patients evaluated in this study, six had PASH, three had PASH/SAPHO overlap and three had PAPASH (Table 1).

A chronic-relapsing course was recorded consistently throughout all investigated cases. Concerning family and personal medical history, none had relatives with PG or HS, but a series of comorbidities was identified, including Crohn’s disease (pat.ID 9,10), ulcerative colitis (pat.ID 8), polycystic ovary syndrome (pat.ID 6,8,10,11), pilonidal cyst (pat.ID 12), psoriasis (pat.ID 4), type II diabetes mellitus (pat.ID 5).

Concerning HS specifically, axillary folds (pat.ID 1–12) were most frequently involved, followed by groin (pat.ID 1–11), anogenital area (pat.ID 2–5,7–12), nuchal region (pat.ID 1,7,9,10) and infra/intermammary folds (pat.ID 5,8,10,11). Mean International Hidradenitis Suppurativa Severity Score System (IHS4) score at first evaluation from these patients was 23.1.

The PG was classified as classic ulcerative in all patients, with associated vegetations in three. The PG onset occurred later than that of HS (mean age at onset, 30.5 vs. 25.9 years, respectively). Affected anatomical sites included the lower limbs (pat.ID 2–4,6,9), trunk (pat.ID 1,2,4,7,11), upper limbs (pat.ID 2,4), peristomal (pat.ID 8,10) and perianal regions (pat.ID 5), with the disease being either localized (pat.ID 3,5,8,10,12), multi-lesional (pat.ID 1,6,7,9,11) or widespread (pat.ID 2,4).

Acne was observed in all patients, manifesting at least 8 years earlier than PG/HS on average (mean age at onset, 16.5 years). The face (pat.ID 1–12) and trunk (pat.ID 1,2,6,8–10,12) were preferentially affected. Arthritis was documented in six patients (mean age at onset, 25.67 years), with wrists and ankle involvement being more prominent in the three cases with PAPASH (pat.ID 1,8,11), and sacroiliac, ankle, and axial/spine involvement in the three with PASH/SAPHO overlap (pat.ID 2,3,7).

Topical therapy with clindamycin gel for HS was prescribed in all cases. Ten patients also received systemic antibiotics and one was given oral isotretinoin. Five patients were administered immunosuppressive/immunomodulating agents, including: systemic corticosteroids (pat.ID 1,4,9), cyclosporine (pat.ID 4,9), dapsone (pat.ID 9,11), intravenous immunoglobulin (pat.ID 7), methotrexate (pat.ID 3,7,10) and colchicine (pat.ID 11). Biologics were also given to all individuals, including adalimumab (pat.ID 1,3–5,7,9–12), infliximab (pat.ID 2,3,5,6,8,10,12), anakinra (pat.ID 1,7,9,11), and ustekinumab (pat.ID 4,7).

### 3.2. Genetic Analyses

We sequenced the whole coding region of *AIM2* gene, comprising 5’- and 3 ‘UTR and a region upstream of the first exon of ~ 800 bp in all patients. In syndromic HS patients, no novel potential causative variants (i.e., private mutations) were identified. Among the polymorphic variants already reported in dbSNP [19], only the rs41264459 SNP (c.−208 A ≥ C) had a higher frequency than the one reported in the general population of European origin. Specifically, six out of twelve syndromic HS patients (PASH = 3; PAPASH = 1; PASH/SAPHO = 2) were heterozygous carriers of the minor C allele. Therefore, the allelic frequency of rs41264459 in syndromic HS sample was 0.25, much higher than that reported in the 1000 G (0.075) and GnomAD (0.094) databases.

The rs41264459 variant is located on the 5′UTR of the *AIM2* gene, within a region in which multiple elements involved in transcriptional regulation were mapped (Figure 1).

We subsequently genotyped the rs41264459 variant in 40 sporadic HS and 23 isolated PG cases. In the HS sample, rs41264459 was found to have an allelic frequency of 0.125 (10 heterozygous subjects), while in PG group the allelic frequency was 0.065 (3 heterozygous subjects).

Given the small number of our cohort, it was not possible to analyze the results with a statistical test; however, these results suggest that this variant may play a role in susceptibility to develop syndromic forms of HS but not to progress to sporadic HS and PG.

## 4. Discussion

The human IFN-inducible protein absent in the melanoma 2 (AIM2) gene encodes a 39-kDa protein that recognizes altered or mis-localized double-stranded DNA (dsDNA). It is a key pathogen sensor that detects the presence of foreign DNA accumulating in the cytosolic compartment during the life cycle of viruses, bacteria, and parasites, thus mediating a protection against the invading pathogens. The AIM2 can also detect damaged DNA, as well as the aberrant presence of DNA released into the cytosol, triggering inflammasome assembly, which in turn leads to both the secretion of bioactive IL-1β, IL-18 and the induction of pyroptosis [4].

Recent studies showed that the detection of self-DNA by AIM2 represents an important factor contributing to diseases associated with perturbation of cellular integrity; indeed, its deregulation can drive the progression of sterile inflammatory disorders including, among others, autoimmune/autoinflammatory skin conditions [20]. For example, in psoriasis-a chronic autoimmune-mediated inflammatory skin disease-cytosolic DNA induces IL-1β release from keratinocytes via an AIM2-dependent inflammasome. This suggests that the specific pro-inflammatory microenvironment of psoriatic skin provides a trigger for AIM2-dependent inflammasome activation, although the source of cytosolic DNA in keratinocytes remains unknown [6].

Syndromic forms of HS can be regarded as autoinflammatory skin diseases that share clinical and pathological features with neutrophilic dermatoses, that encompass a wide spectrum of conditions hallmarked by an accumulation of neutrophils in the skin and, rarely, internal organs. The autoinflammatory nature of these conditions has been confirmed by the discovery of mutations in *PSTPIP1* (proline–serine–threonine phosphatase-interacting protein 1), a gene involved in innate immunity. The latter lead to reduced inhibition of the inflammasome and consequent activation of caspase-1, which in turn results in an increased production of IL-1β, driving neutrophil-mediated inflammation [21].

To date, the NLR family pyrin domain containing 3 (NALP3), encoded by *NLRP3* gene, is the most studied inflammasome as it has been found to be associated to various inflammatory, metabolic, and neurological diseases. Just like AIM2, the NLRP3 inflammasome promotes an inflammatory cell signaling ultimately resulting in the release of IL-1β and pyroptosis [3]. Marzano et al. described a missense variant of uncertain significance (VUS) with pathogenic evidence in *NLRP3* gene (NM_001243133.2:c.2107 C > A; p.Gln703Lys) in a PASH patient, reinforcing the potential causative role of dysregulated innate immunity and IL-1β-dependent autoinflammation in this spectrum of diseases [22]. Indeed, other potentially pathogenic variants in autoinflammatory genes, including *MEFV*, *NOD2*, *PSTPIP2* and *NLRC4*, have been reported in HS syndromic forms [14,23,24]. All these protein-coding genes can promote an increased IL-1β release, thus initiating and amplifying a wide variety of effects linked to innate immunity, tissue damage and autoinflammation [23].

Given the prominence of the IL-1β pathway in the complex pathogenetic scenario of HS and its syndromic forms, it is likely that AIM2 also plays a role. Therefore, we analyzed the whole coding region of *AIM2* gene including its promoter, 5′UTR and 3′UTR regions. Six out of twelve syndromic HS patients (PASH = 3; PAPASH = 1; PASH/SAPHO = 2) carried a heterozygous variant c.−208 A ≥ C (rs41264459) with a MAF of 0.25, which is much higher than that reported in reference sets (0.075 and 0.094 for the 1000 G and GnomAD, respectively). This variant is located at the promoter region of the *AIM2* gene; therefore, it might have an impact on gene expression levels. The same heterozygous variant was previously found in ten patients diagnosed with PFAPA (Periodic fever with aphthous stomatitis, pharyngitis, and adenitis) syndrome, the most common autoinflammatory disorder in childhood, albeit without reaching statistical significance [25].

Likewise, given our small sample size, it was not possible to analyze the results with an appropriate statistical test, but it is plausible to assume that this variant may play a role in the pathogenesis of HS syndromic forms. Furthermore, this variant was found at a lower allelic frequency in sporadic HS and isolated PG (0.125 and 0.065, respectively), leading to the hypothesis that it may play a role in susceptibility to syndromic forms of HS but not to sporadic HS and PG. Functional validation of this variant was not performed due to the unavailability of biological samples (i.e., skin biopsies) for RNA extraction. However, it is possible that non-coding regions, epigenetic modifications as well as somatic variations also affect gene expression, resulting in diverse, context-dependent responses, that may be difficult to assess and validate from a functional perspective.

Based on the role of AIM2 in autoinflammatory conditions, these preliminary results could pave the way to further investigations on this gene, evaluating its epigenetic patterns and its expression in the skin and in different cell types of patients with syndromic HS.

## 5. Limitations and Strengths

Our study presents two limitations: the first one is the small sample size, which, albeit limited, represents one of the largest syndromic HS cohorts analyzed up to now; the second one is related to the lack of validation, which will be the subject of further research.

## Figures and Tables

**Figure 1 vaccines-11-00162-f001:**
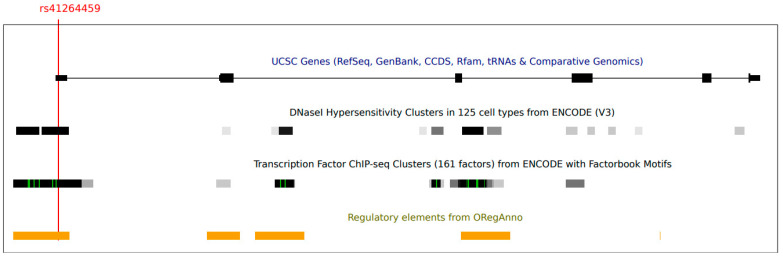
A screenshot from the UCSC genome browser (http://genome.ucsc.edu/, GRCh37/hg19, accessed on 1 December 2022). The panel shows the AIM2 gene and the informative tracks showing DNase hypersensitive areas (from ENCODE), regions of transcription factor binding (from ENCODE), literature-curated regulatory regions and transcription factor binding sites (from OregAnno). The position of variant rs41264459 is also reported.

**Table 1 vaccines-11-00162-t001:** Clinical features of patients with syndromic hidradenitis suppurativa. CR, complete response; PR, partial response; NR, no response; BBS, bowel bypass syndrome; BP, bullous pemphigoid; DM, diabetes mellitus; FMF, familial Mediterranean fever; PCOS, polycystic ovarian syndrome; PPPP, palmoplantar pustular psoriasis; PAPASH, pyogenic arthritism pyoderma gangrenosum, acne and hidradenitis suppurativa; PASH, pyoderma gangrenosum, acne and hidradenitis suppurativa; SAPHO, synovitis, acne, hyperostosis, and osteitis.

ID	Sex	Diagnosis	Comorbidities	Hidradenitis suppurativa			Pyoderma gangrenosum			Acne		Treatments	Follow-up
				Age at onset (years)	Involved sites	IHS4	Age at onset (years)	Involved sites	PG Score	Age at onset (years)	Involved sites		
1	M	PAPASH	pyogenic arthritis	14	axillae, groins, back, nuchal region, scalp	30	18	trunk	multi-lesional	13	face, back	rifampicin, doxycycline, azitromycine, lymecycline, clindamycin, adalimumab, anakinra, prednisone	CR
2	M	PASH/SAPHO	SAPHO	43	axillae, groins, anogenital area	27	43	lower limbs, trunk, upper limbs	disseminated	17	face, back	clindamycin, infliximab	CR
3	M	PASH/SAPHO	SAPHO	42	axillae, groins, anogenital area	19	46	lower limbs	localized	16	face	clindamycin, methotrexate, infliximab, adalimumab	CR
4	M	PASH	Obesity, BBS, osteoporosis, PPPP	36	axillae, groins, anogenital area	24	36	lower limbs, trunk, upper limbs	disseminated	34	face	trimetoprim-sulfametoxazole, azitromycine, cyclosporine, prednisone, adalimumab, ustekinumab,	CR
5	F	PASH	Obesity, BBS, myocardial infarction, type II DM, BP	51	axillae, groins, submammary folds, anogenital area	23	51	perianal region	localized	15	face	rifampicin, doxycycline, infliximab, adalimumab	CR
6	F	PASH	PCOS	15	Axillae, groin	20	24	Lower limbs	multi-lesional	14	Face, back	infliximab	CR
7	F	PASH/SAPHO	SAPHO	15	axillae, groins, anogenital region, nuchal region	26	15	trunk	multi-lesional	13	face	rifampicin, doxycycline, azitromycine, lymecycline, clindamycin, intravenous immunoglobulin, methotrexate, isotretinoin, adalimumab, anakinra, ustekinumab, secukinumab	NR
8	F	PAPASH	Ulcerative colitis, PCOS, pyogenic arthritis	28	submammary folds, groins, anogenital area	22	36	peristomal	localized	15	face, back	doxycycline, infliximab, surgery for HS	CR
9	M	PASH	Crohn disease, osteoporosis	18	axillae, groins, anogenital area, nuchal region	23	19	lower limbs	multi-lesional	16	face, back	azitromycine, trimetoprim-sulfametoxazole, ceftriaxone, ciprofloxacine, lymecycline, rifampicin, amoxicillin clavulanate, methylprednisolone, dapsone, cyclosporine, adalimumab, anakinra,	PR
10	F	PASH	Crohn disease, coeliac disease, myasthenia gravis, PCOS	18	axillae, nuchal region, submammary folds, groins, anogenital area	19	44	peristomal	localized	17	face, back	clindamycin, methotrexate, infliximab, adalimumab, surgery for HS	CR
11	F	PAPASH	FMF, PCOS, pyogenic arthritis	16	axillae, groins, anogenital area, intermammary and submammary folds	20	16	trunk	multi-lesional	14	face	azitromycine, doxycycline, dapsone, adalimumab, anakinra, colchicine	CR
12	M	PASH	Pilonidal cyst	15	axillae and anogenital	21	18	lower back	localized	14	Face, back	Azithromycin, infliximab, adalimumab	PR/CR

## Data Availability

The data are contained within the article.

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
