# Peer review of "Autoinflammation in Syndromic Hidradenitis Suppurativa: The Role of AIM2"

_vaccines, 2023, doi:10.3390/vaccines11010162_

Round 1

Reviewer 1 Report

It was a pleasure for me to read this interesting paper. The purpose, focusing on inflammasome and its activation pathways, has great scientific relevance at present time and the article is well structured. The weakness of small sample size was already declared by AA. In my opinion the manuscript is already suitable for publication as it is.

Author Response

We would like to thank the Reviewer for the positive comments and the appreciation of our work.

Reviewer 2 Report

Syndromic HS represents a group of syndromes with clinical signs exhibiting an autoinflammatory background as common denominator. The syndromes are rare and include Hs as one of their major signs. While several research groups concentrate their work on the elucidation of HS etiopathogenesis, there is only a few information on the pathology of syndromic HS and especially any common molecular background.

Since autoinflammatory diseases are likely to at least partially be interferonopathies, it is consequent that the authors, experts in the field of syndromic HS, looked for the expression of the human IFN-inducible protein absent in melanoma 2 (AIM2) gene in these diseases.

The data presented are preliminary and a small number of patients have been included in the study, but the data are valuable for the presentation of the concept and as background for further larger studies.

Author Response

(The authors gave the same response as above.)

Reviewer 3 Report

Table 1 format: please check the treatment/follow up column. The PDF format I reviewed did not align.

Line 139-line 161: when using parenthesis, make it clear the number is referring to the patient ID in the table. It may be confusing if it may reflect the number of patients with the condition.

line 144. clarify "mean". I suppose the mean score of at first evaluation from these patients.

Line 147: what 30.5 vs 25.9 reflected? The average age at onset of these patients? clarify.

line 163: rs41264459 variant - explain in the introduction. 

Figure 1: improve the visuals. Not clear.

Line 189-191/246-248. It is not clear. Rewrite. It is easier to understand the language in the abstract

.
